# Addressing the polypharmacy challenge in older people with multimorbidity (APOLLO-MM): study protocol for an in-depth ethnographic case study in primary care

Deborah Swinglehurst, Nina Fudge

Centre for Primary Care and Public Health, Blizard Institute, Barts and The London School of Medicine and Dentistry, Queen Mary University of London, London, UK

**Correspondence to**
Professor Deborah Swinglehurst;
d.swinglehurst@qmul.ac.uk

## ABSTRACT

**Introduction** Polypharmacy is on the rise. It is burdensome for patients and is a common source of error and adverse drug reactions, especially among older adults. Health policy advises clinicians to practice *medicines optimisation*—a person-centred approach to safe, effective medicines use. There has been little research exploring older patients' perspectives and priorities around medicines-taking or their actual practices of fitting medicines into their daily lives and how these are shaped by the wider context of healthcare.

**Methods and analysis** We will conduct an in-depth multisite ethnographic case study. The study is based in seven clinical sites (three general practices and four community pharmacies) and includes longitudinal ethnographic follow-up of older adults, organisational ethnography and participatory methods. Main data sources include field notes of observations in the home and clinical settings; interviews with patients and professionals; cultural probe activities; video recordings of clinical consultations and interprofessional talk; documents. Our analysis will illuminate the everyday practices of polypharmacy from a range of lay and professional perspectives; the institutional contexts within which these practices play out and the sense-making work that sustains—or challenges—these practices. Our research will adopt a 'practice theory' lens, drawing on the sociology of organisational routines and other relevant social theory guided by ongoing iterative data analysis.

**Ethics approval** The study has HRA approval and received a favourable ethical opinion from the Leeds West Research Ethics Committee (IRAS project ID: 205517; REC reference 16/YH/0462).

**Dissemination** Aside from academic outputs, our findings will inform the development of recommendations for practice and policy including an interactive e-learning resource. We will also work with service users to co-design patient/public engagement resources.

## Strengths and limitations of this study

► Gathers rich qualitative data across community settings (the home; general practice; pharmacy) with a focus on how practices unfold on the ground and how different actors make sense of polypharmacy.
► Brings together analysis of patients' lived experience with analysis of organisational routines and professional practices.
► Uses innovative qualitative methodologies which generate more sophisticated insights than is possible with conventional approaches.
► Prioritises depth of analysis over breadth, increasing richness of understanding but limiting generalisability of findings.

adults are dispensed 10 or more drugs (this tripled over the same time period).[1] There is growing concern that polypharmacy is one example of 'medical overuse', causing unnecessary burden, iatrogenic harm to patients and costly waste to health systems. Up to half of medicines prescribed for long-term conditions are unused.[2] This is often conceptualised as a problem of poor adherence[3–5] or as a failure of shared decision-making in the consultation.[6 7] The problem is assumed to be located within individuals' behaviours, with relatively little attention paid to the relationship between individuals and the wider social, organisational and institutional context within which their medicines practices play out.

Polypharmacy is often attributed to the ageing population; 65% of people aged 65–84, and over 80% of people aged 85+ experience multimorbidity.[8] But ageing per se is only one of many potentially relevant factors. For example, evidence-based guidance which informs prescribing is typically organised around a 'single-disease' model

## INTRODUCTION

Polypharmacy is increasing. One Scottish study showed that about 20% of adults are dispensed five or more drugs (this doubled between 1995 and 2010) and about 6% of

**BMJ**

and may compound rather than alleviate polypharmacy. One critic has described evidence-based medicine as a perverse 'monument to bias', built on its systematic exclusion of complex patients with multimorbidity from the trials that inform it.[9] Arguably, polypharmacy in the context of multimorbidity is rarely, if ever, evidence based, even when a clear 'evidence-based' argument can be made for individual items.[10] Achieving *medicines optimisation*—a person-centred approach to safe, effective medicines use[11–13]—is fraught with difficulty when the institutional context privileges the single-disease model and is increasingly concerned with the application of abstract, generalisable rules. This is further complicated by the increasing emphasis on what has been called 'risk factorology' and the quest to eliminate risk of disease altogether.[14] The societal context is increasingly one in which 'every responsible and rational citizen is expected to actively seek out and eliminate all possible risks to their future health and to consume medical technologies in order to achieve this aim'.[15] Pay-for-performance initiatives which embrace (and arguably constitute) aspirations to eliminate risk and uncertainty may also drive polypharmacy.

A recent study which examined temporal trends between 2000 and 2015 in the use of tests and investigations in UK primary care found that age-adjusted and sex-adjusted use of tests and investigations increased by a staggering 8.5% *annually*.[16] Although prevalence studies do not offer insight into the drivers of change, it seems reasonable to conclude that the parallel and dramatic rise in investigations and polypharmacy over a similar time period may be related phenomena. Privileging priorities for reducing risk of disease may contribute paradoxically to new drug-related risks—risks which increase with increasing age and with increasing numbers of medications.[17] 'Problematic' polypharmacy[13] arises insidiously, despite (and sometimes even because of) clinicians and patients striving to achieve best practice and acting with the very best of intentions. The elusive boundary between 'just enough medicine' and 'too much medicine' is unstable, contingent and socially constructed. It involves clinical judgement in the context of inadequate evidence and considerable uncertainty. Sinnott has referred to the work of negotiating this boundary as a process of 'satisficing'.[18] Doctors report settling for management that is satisfactory and sufficient given the particular circumstances of a patient—a combination of relaxing nationally recommended targets, compromise, hunch, guesswork and maintaining the status quo when patients appear stable.[18]

Polypharmacy presents a complex challenge for clinicians and health systems. It arises at the interface of patients, clinicians, diseases (and risk thereof) and involves a tangle of biological, cultural, technological, economic and sociopolitical dimensions. It is a 'wicked problem'.[19] One feature of wicked problems is their resistance to adequate definition. Polypharmacy tends to be defined numerically, with one recent systematic review identifying 138 different definitions.[20] But simple numerical definitions conceal much of the complexity that the polypharmacy concept evokes.[17] Rittel states that 'the formulation of a wicked problem *is* the problem'; wicked problems are ill-defined, not amenable to a quick-fix 'solutions' but at best '*resolved, or rather re-solved, over and over again*'. Wicked problems coalesce around important moral issues whereby there are acknowledged discrepancies between the state of affairs as it *is* and the state as it *ought* to be.[19] Ethnographic approaches are especially good at investigating complex issues such as polypharmacy and shedding light on ill-defined but troubling phenomena.[21] In this study, we will adopt an ethnographic approach and apply a range of innovative methods which focus on meanings, experiences and practices, as we encounter them in the lives of patients and professionals. This approach embraces polypharmacy as a wicked problem, offering 'new ways of looking' and thereby opening up potential new avenues for addressing polypharmacy in practice and policy.

## AIM

The aim of the APOLLO-MM project is to improve patient care by producing 'practice-based' evidence[22] to inform medicines optimisation. We will do this by:

▶ Examining patients' experiences of polypharmacy to discern elements of high-quality 'medicines work' and those elements that might lead to harmful, wasteful or unnecessary polypharmacy.

▶ Improving understanding of how polypharmacy is sustained and/or challenged within and between lay and professional networks.

▶ Analysing consultations that include talk about medicines (including but not limited to formal 'medication review') to understand how medicine optimisation is negotiated in practice.

▶ Using participatory methods to elicit professional dialogue around polypharmacy and offer opportunities to discern the elements of good practice in medicines optimisation.

▶ Working with patients and professionals in a process of co-design to develop e-learning materials, recommendations for practice and public/patient engagement resources.

## RESEARCH QUESTIONS

1. What is the patient experience of polypharmacy in multimorbidity?
2. What does polypharmacy mean for patients and carers?
   a. How are patients' lives with multimorbidity shaped by practices of managing medicines?
   b. How is the managing of medicines shaped by contingencies of living with multimorbidity?
3. How do the practices of patients/carers/health professionals and wider systems of care support 'appropriate' polypharmacy or challenge 'inappropriate' polyphar-

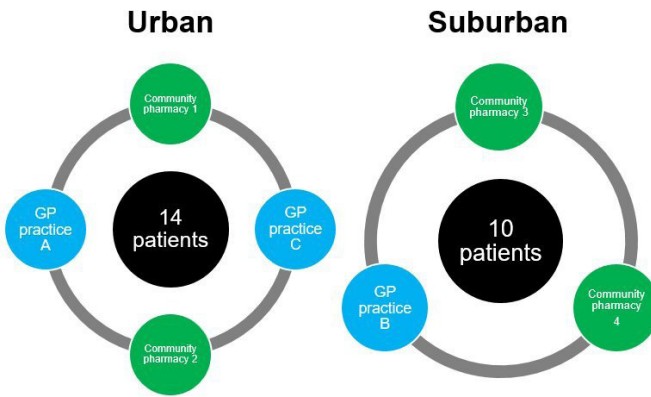

**Figure 1** Study sites and research clusters. GP, general practitoner.

macy and with what consequences for the quality and safety of care?

4. How can insights from longitudinal patient case studies inform improvements in professional practices, service delivery and public awareness of polypharmacy?

## METHODS
### Setting and context
The study is a multicentre ethnographic case study involving seven clinical sites across two research clusters in England (one urban and one suburban cluster). The urban cluster comprises two National Health Service (NHS) general practitioner (GP) practices (Site A and Site C) and two independent community pharmacies located in an area of high socioeconomic deprivation within a large city, serving a population of diverse ethnicity comprising approximately 40% Asian patients. The suburban cluster comprises one NHS GP practice (Site B) and two independent community pharmacies, located in a small, relatively prosperous city serving a population which is 85% white Caucasian. The research clusters represent contrasting demographics; the urban cluster serves patient populations within the lowest (Site C) and second lowest (Site A) decile of the English Index of multiple deprivation score,[23] and the suburban cluster serves a population within the least deprived decile.

The GP practices were selected following expressions of interest from over 40 GP practices within the North Thames Clinical Research Network area, suggesting that the project resonates with the concerns of GPs on the ground.

We have recruited 24 patients, 14 in the urban cluster (seven each from Sites A and C) and 10 from our suburban cluster. The work of patient follow-up and organisational ethnography has begun. The GP practices and community pharmacies in the study share responsibility for prescribing and dispensing medicines to the study patients, allowing for multiple perspectives of polypharmacy. To date, we have recruited 40 professional participants for interviews, filming of consultations and participation in research workshops (see the Study design, methodology and approach to analysis section).

The relationship between the patients, GP practices and community pharmacies is shown in figure 1.

### Selection of organisational cases
Drawing on Stakes approach to organisational case study our selection of GP sites primarily reflects (1) *opportunity for learning*, rather than concerns about representativeness[24] and (2) diversity of geographical setting and population demographics. Our concern is to conduct an in-depth study of *particular* practices within these sites, adopting an interpretive approach informed by relevant social theory. We will create generalisable theoretical and conceptual abstractions (novel concepts to 'think with') rather than generalisable statistical abstractions. The ethnographic approach relies on the interest shown in the study by the potential participants, the willingness of staff at all levels of the organisation to allow meaningful access to the research team (eg, shadowing, observations, formal and informal interviews; access to relevant documents) and enthusiasm for the participatory element of the study within the context of already heavy workloads (see the Study design, methodology and approach to analysis section). In addition, we considered a range of pragmatic concerns relating to feasibility, location and likelihood of building sustainable working relationships for the duration of the study. The table 1 shows some of the characteristics of our recruited practices.

### Sampling of patient participants
We have recruited patients aged 65 or over and prescribed 10 or more regular (repeat) items of medication, this being regarded a pragmatic marker of 'high-risk polypharmacy'.[13] Where patients were prescribed an item in more than one dose (eg, warfarin), this was counted as a single item. Medical devices and hosiery were excluded, but medicines in all forms (eg, oral, inhalers, injectables, topical creams, etc) were included. We adopted a purposive sampling approach, aiming for a maximum diversity sample across a range of attributes: age (ranging from 65 to 94); gender (11 men; 13 women); living circumstances (16 live alone of whom five in sheltered housing; eight with partners/family; four participants are housebound); socioeconomic status (home ownership and previous occupation as proxy indicators) and number of medication items.[10–17] All patients have two or more comorbidities. Twenty-one out of 24 patients are white Caucasian. We have excluded patients unable to speak adequate English, which means that sampling in our urban cluster is not representative of the local population. We are addressing this limitation in another study.[25]

Recruitment took place in general practice. GPs and nurses introduced the study to potential participants, issued participant information leaflets and sought verbal consent from patients to be contacted by a researcher (n=63). Thirty-six patients agreed to a home visit to discuss the study in more detail and 24 consented to take part. Patients lacking capacity to consent were eligible for inclusion on the identification of a personal consultee.

**Table 1** Characteristics of recruited GP practices

| | Site A | Site B | Site C |
|---|---|---|---|
| Cluster | Urban | Suburban | Urban |
| Patient population | c 11 000 | c 13 000 | c 14 000 |
| GPs | 5 partners; 5 salaried | 4 partners; 6 salaried | 8 partners; 3 salaried |
| Onsite clinical pharmacist at recruitment | No | No | No |
| Onsite clinical pharmacist for part of project duration | Yes | Yes | Yes |
| GP training practice | Yes | Yes | Yes |
| Deprivation* | Second most deprived decile | Least deprived decile | Most deprived decile |
| Ethnic diversity (estimated proportion non-white ethnic groups)* | Over 50% (over 40% of practice population are Asian) | 14% | Over 60% (over 40% of practice population are Asian) |

*Details from National General Practice Profiles produced by Public Health England Data Science.
GP, general practitioner.

A consultee is someone who, by virtue of their existing relationship with the potential participant, can advise the researcher about their participation in the project. One patient (with dysphasia following a stroke) has a personal consultee, their next of kin, who is their primary caregiver. We were unable to recruit any patients with dementia; we are addressing this using a different recruitment strategy in a parallel study which focusses specifically on patients with dementia[25]

### Study design, methodology and approach to analysis

The figure 2 shows the methods and data sources for the study, which are designed to elicit and explore patient experiences and practices; professional experiences and practices and professional dilemmas. These are described in detail in the following sections.

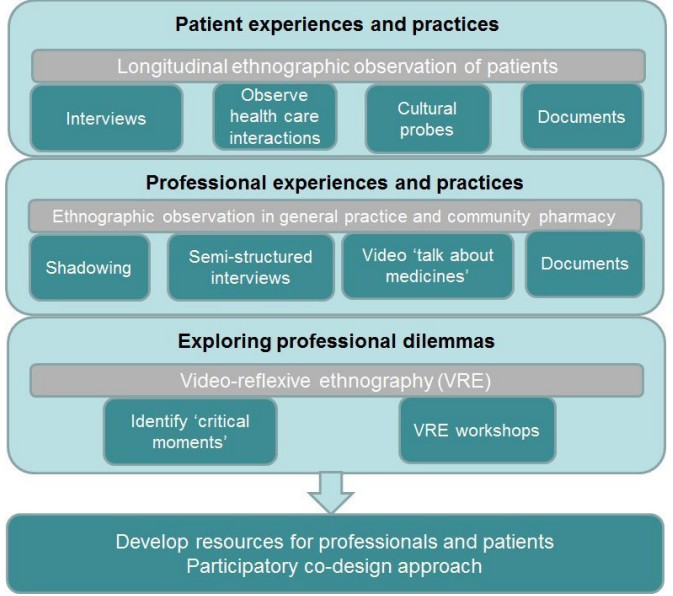

**Figure 2** Methods and data sources.

### Patient experiences and practices

We will follow patient participants ethnographically over 18–24 months, observing them at home and accompanying them to selected healthcare interactions in which they anticipate talk about their medicines (eg, at the GP surgery, pharmacy or hospital). We are conducting interviews with patients to elicit the story of their lives since they were first advised to take medicines, adapting Wengraf's Biographic-Narrative Interpretive Method.[26] This approach utilises a single question to induce narrative and invites participants to speak uninterrupted until their story is complete. After a short break (10–15 min), the researcher presents a series of further questions based on 'cue phrases' (the patient's own words) in the order the patient introduced them into their opening narrative, to encourage further in-depth narratives (particular incident narratives). On a separate occasion, we return to the patient's home to conduct an in-depth interview guided by a topic guide with a more explicit focus on medicines practices. We contact participants by telephone or home visit at approximately 4–8 weekly intervals to learn about their ongoing experiences and to identify occasions when we may accompany them to a healthcare interaction or video record a consultation. Every time we meet our participants, we take detailed ethnographic field notes, add to our reflexive journal and gather relevant documents (eg, discharge summary or lists of medication). This data enables us to build detailed patient case studies as we follow their lived experience and engage them in informal 'event-based' interviews exploring their care experiences.

A subset of patients are completing cultural probe activities as a way of depicting their lives. Cultural probes originated in the world of design as a way of tapping into peoples' creativity and offering research participants opportunity to engage in a research-related activity that is subjective, deliberately ambiguous, open

to interpretation, invites imagination and stimulates conversation.[27 28] The probes consist of a set of activities and a range of materials (eg, camera, pens, cards, booklets) with simple instructions which offer opportunity for improvisation. They encourage participants to think about their medicines and bodies in new ways and reimagine possibilities for how they might interact with health professionals. The creative outputs are used to stimulate an informal discussion between researcher and participant. Participants may invite a family member or researcher to assist if they wish. Cultural probes have been used successfully to incorporate older people's ideas into the design of social communities[27], and to illuminate the daily living needs of older people to inform co-production of assistive living technologies,[29 30] although it is recognised that some people may be unable to engage with them for a range of physical, emotional and social reasons.[29]

### Professional experiences and practices

We will conduct ethnographic observations in three GP and four community pharmacy practices (see figure 1). Our interest is in understanding how health professionals conduct and make sense of their day-to-day work of prescribing, dispensing, negotiating, organising medicines for patients affected by polypharmacy. We will observe key routines such as preparation and organisation of multicompartment compliance aids (MCAs or 'dosette boxes'); repeat prescribing and repeat dispensing.

We will closely observe individuals at work (shadowing and asking participants to describe what they are doing as they do it to encourage the articulation of tacit knowledge); organisational routines and how they are constituted; the wider institutional context and documents relevant to the work we observe (eg, prescriptions, guidelines). We will also interview key professional participants across our sites with a focus on accounts of their working practices. We will hone in on consultations between clinicians and our study patients to examine healthcare interactions in detail by video recording consultations in which our participants anticipate 'talk about medicines'. The polypharmacy context is a very particular context for these interactions and we are interested to learn how polypharmacy is sustained, challenged, talked about and 'brought about' in these interactions.

Organisational routines are conceptualised as 'repetitive recognisable patterns of interdependent actions by multiple actors'[31 32] and can be regarded as sites of stability and as sites of potential for innovation and change.[33] When participants engage in routines they enact tacit knowledge and are involved in complex patterns of collaboration—involving people, artefacts, technologies—to get work done. Routines can be embodied, embraced, resisted and enacted with varying degrees of creativity and improvisation and varying attention to the particularities of local context.[34] Pentland and Feldman have conceptualised routines as being composed of:

▶ Ostensive routines: the generalised 'rules' or organisational scripts that guide the routine, of which there may be many instantiations.
▶ Performative routines: what people *actually do in practice* as they engage in the routine. Each instantiation of the routine is a unique performance.

Ostensive routines may be accessed through peoples' accounts of the work they do, while observation of repeated rounds of a routine gives insight into the many versions of the performative routine and the relationship between the performative and the ostensive aspects of the routines. When people engage in routines, they select from a repertoire of possible actions (enacting possibilities around which there are explicit or implicit constraints). Different people bring different knowledge, different expertise, different assumptions and different personal and organisational narratives to the enactment of the routine. An understanding of the patterns of possibility and constraints offers insights into wider organisation principles and values, and to an appreciation of complex organisational practices, such as how power circulates and how learning is done. The dynamic interplay between the ostensive and performative routines and the artefacts which are drawn on to support these routines can transform what has traditionally been regarded as mundane and ordinary organisational life (eg, the processing of a repeat prescription) into opportunity for 'thick description'[35] of the nature of organisational life and the wider context within which work is being done.[32 36] Continuities and discontinuities in the relationship between ostensive and performative routines may become manifest as puzzles, conflicts, moments of perplexity or tensions.

Recent literature on organisational routines has pointed to the need to pay greater attention to the relational nature of 'self' and agency in the performance of routines and in the dynamic processes that address *how routines become collective accomplishments* with recognisable shared collective characteristics (ie, how the 'ostensive' aspect of routines is constituted or brought about).[37 38] One approach which offers promise is to locate the study of routines within emerging 'communication as constitutive of organisation (CCO)' scholarship,[39–42] where communication is itself framed as performance (through conversation and text dialectic). Routines are then framed as communicatively constituted performances, there is greater emphasis placed on performativity, embodiment, dialogic inter-relationships and interest in 'strings of associations' that link actors (human and non-human) as routines unfold.[43] The nuance of communication gains ground—the emergence of organisational storylines, for example.[44–46] This approach aligns with our interest in patient narratives and our orientation to 'talk about medicines' as co-constructed accomplishment within the consultation.

### Video reflexive ethnography (VRE)

VRE is an innovative approach to quality and safety research in healthcare[47–49] which encourages groups

of professionals to discuss and reflect on their working practice. In this study, selected video recorded footage of 'medicines talk' in our study patients' consultations will be shown to small groups of professionals across our research sites. VRE offers an opportunity for participants to negotiate shared meanings in their practice, engage with the complexity of their everyday work, and make visible—in new ways—aspects of their practice which have become habituated, taken for granted and 'invisible'.[49] It is a participatory approach in which researchers and practitioners work together to illuminate this complexity and foster dialogue that might not otherwise occur.[50] The footage acts as a catalyst, making implicit knowledge explicit through a process of reflection-on-action which renders the work (in this case, the practical work of 'medicines optimisation') understandable in new ways and open to new ways of re-imagining practice.[51] Importantly, the discussion may go beyond what is visible in the footage, the footage acting as a prompt to inter-professional exchange and peer-led discussion of professional concerns, dilemmas and opportunities. There is opportunity for 'exnovation' or innovation from *within* practice[48] and the cultivation of meta-discourses of practice[50] which can reframe everyday practices (eg, the 'medication review') by drawing out the systemic implications of what specific people know, do and say. By video recording the discussion, this meta-discourse can be teased out and worked with across VRE sessions, in ways which enable practitioners to iteratively reframe their work and negotiate new possibilities for action. Importantly, the collaborative process of co-constructing this meta-discourse in the VRE context may itself be a rehearsal space for new ways of relating and collaborating in the very doing of the work that is under scrutiny.

### Participatory co-design
The findings from the research will feed into a process of participatory co-design, working in collaboration with both patients and professionals to design e-learning materials for professional education and patient engagement resources for patients. In this work, we will adapt processes of experience-based co-design[52 53] and 'future groups',[54] the latter being a reinterpretation of focus group methods which uses ideation tools and encourages speculation to imagine alternative approaches to care. We will collaborate with a design researcher to develop our research materials and findings to create bespoke ideation tools and props to facilitate imaginative participation and move towards exploring future alternatives to care prompted by the stories, data and findings from the earlier research.

### Integrating analysis across diverse data sets
The project is situated within a broad 'practice theory' orientation and adopts a social constructionist orientation. Practice theory encourages an understanding of work as an accomplishment and fosters a curiosity about what is accomplished by whom and how. The organisation

(eg, the GP practice or pharmacy) is understood as both the *site* and the *result* of work activities and the role of the interpretive researcher is to 'zoom in' on the details of particular practices and to 'zoom out' to discern the relationships and connections between people, routines, artefacts, spaces and technologies, for example.[55 56] As our analysis progresses, we will, in addition, draw on relevant social theory (eg, Burden of Treatment theory[57 58]) to explicate our findings.

This detailed ethnography will extend our previous research on repeat prescribing in general practice settings in four ways.[36] First, the institutional context has changed (one example being the implementation of the Electronic Prescribing Service, another being the incorporation of clinical pharmacists within some GP settings). Second, our research extends beyond general practice into community pharmacy settings and into peoples' homes and opens up a wider range of vantage points from which to observe the phenomenon. We are focussing our observations around the phenomenon of polypharmacy, recognising that polypharmacy emerges out of an arrangement of practices and is not a discrete entity that can be easily observed without appreciation of a wider social context. Finally, our research includes a commitment to 'working with' both health professionals and patients in a process of co-design to produce outputs which we hope will reflect what really matters to staff and patients on the ground.

## PATIENT AND PUBLIC INVOLVEMENT
We have a project advisory group of 11 members including academics, health professionals, representation from Age UK, two patient members and a lay chair. Patients have been involved in the proposal development, design of participant materials and project website (www.polypharmacy.org.uk), our application for ethical approval, the project launch event, piloting of interviews, study design and conduct. We have an online patient panel of five members.

## PROJECT MANAGEMENT AND GOVERNANCE
Our project is overseen by an expert advisory group (see above) and we report annually to the Heath Research Authority and our funder (National Institute of Health Research).

Site recruitment began in March 2017. Patient recruitment began January 2018 and was completed October 2018. Project funding ceases on 28/2/2021.

## ETHICS AND DISSEMINATION
The project has ethics approval (IRAS project ID: 205517; REC reference 16/YH/0462).

Our findings will inform the following key outputs.
1. For clinicians: key recommendations for practice; interactive online e-learning resources.

2. For professional bodies: summary of findings to inform professional guidance.
3. For patients and carers: co-designed resources to inform patient and public understanding of polypharmacy, inform decision-making and improve consultations with health professionals.
4. For policymakers: a summary report of key findings.
5. Academics: research publications and conference presentations.

**Acknowledgements** We would like to thank the staff and patients who have agreed to participate in this study. We would also like to thank members of our Expert Advisory Group and our patient panel for their valuable advice.

**Contributors** DS designed the study and secured funding. DS and NF have both been involved in fieldwork and have both contributed to writing this paper.

**Funding statement** This article presents independent research funded by the National Institute for Health Research (NIHR) through a Clinician Scientist Award (DS). In addition NF was (in part) supported by the National Institute for Health Research (NIHR) Collaboration for Leadership in Applied Health Research and Care (CLAHRC) North Thames at Bart's Health NHS Trust. The views expressed are those of the author(s) and not necessarily those of the NHS, the NIHR or the Department of Health and Social Care.

**Competing interests** None declared.

**Patient consent for publication** Not required.

**Provenance and peer review** Not commissioned; externally peer reviewed.

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
