## [Reviewer comments · BMJ Open]

ARTICLE DETAILS

TITLE (PROVISIONAL)	Addressing the Polypharmacy Challenge in Older People with Multimorbidity (APOLLO-MM): study protocol for an in-depth ethnographic case study in primary care
AUTHORS	Swinglehurst, Deborah; Fudge, Nina

VERSION 1 – REVIEW

REVIEWER	P.M. Mannucci, MD IRCCS Ca' Granda Maggiore Policlinico Hospital Foundation and University of Milan
REVIEW RETURNED	12-Jun-2019

GENERAL COMMENTS	When I accepted to review this article on a protocol related to polypharmacy in older people with multimorbidities I thought to be competent enough for this topic, as a clinician with some specific competence on clinical pharmacology. Yet, reading this protocol, I realized that in it there are a lot of technical aspects related to ethnography that are a little bit out of my brain. With this preamble, I read easily this protocol and I found interesting and meaningful the described approach and the information supplied. However, you also need a more expert reviewer with more specific experience on public health and ethnography.
---

REVIEWER	Wilma Knol University Medical Center Utrecht, the Netherlands
REVIEW RETURNED	20-Jun-2019

GENERAL COMMENTS	First of all I would like to convey my appreciation towards the originality of the qualitative methodologies in this study protocol. This is a comprehensive and well written protocol paper. It has been recognized in multiple previous studies that optimisation of medicines in older patients with polypharmacy is a major challenge. I think there is a need for this type of studies to gain better insight in the complexity of polypharmacy from different perspectives. I recommend it for publication. I would advise the authors to kindly look into the following few points: Introduction: The introduction is quite long. The more general information about polypharmacy is well known. I suggest to shorten the introduction and focus on issues in polypharmacy that have not been studied yet or could be studied using innovative methodologies providing new insights and explain the strategy to solve the issues that need to be further explored. Methods: The total number of participants seems low to me, especially representation of non-white Caucasians. It was mentioned that a purposive sampling approach was used, but information about data saturation is lacking. A more
--

	detailed data analysis section is missing in this protocol paper. I assume that it is difficult as this protocol consists of different parts. Please elaborate.
--	---

VERSION 1 – AUTHOR RESPONSE

Peer Reviewer 2

Introduction: The introduction is quite long. The more general information about polypharmacy is well known. I suggest to shorten the introduction and focus on issues in polypharmacy that have not been studied yet or could be studied using innovative methodologies providing new insights and explain the strategy to solve the issues that need to be further explored.

Many thanks for these suggestions. We have recrafted the introduction, removing some of the well-known clinical knowledge and limiting this to what we regard as essential context. We have retained references to ‘adherence’ in order to set out (albeit briefly) the need for a more critical approach to adherence. We have retained the sections which set out the tension between the contemporary institutional context (with its focus on single diseases, risk factors, population derived evidence and generalizable rules) and the call for a patient-centred approach. We have removed reference to the UK QOF as we realise the readership is international. We have improved the section on polypharmacy as a ‘wicked problem’ and have more clearly situated our research and its innovative methodologies as a way of investigating polypharmacy as a complex, wicked problem and thereby opening up new possibilities for addressing it.

We have reduced the length of the introduction from 869 to 806 words. This is a modest reduction, but we feel we have succeeded in shifting the focus towards issues that are most relevant to our particular study as Reviewer 2 suggested.

Methods: The total number of participants seems low to me, especially representation of non-white Caucasians.

We have recruited 24 patient participants. This sample size was agreed with our expert advisory group and is relatively large for in-depth ethnographic work of this kind. The research is very detailed, and involves multiple occasions of interviewing, home visiting, telephone contact and cultural probe activities with these patient participants over 18-24 months, as well as some accompanying visits to health care appointments and filming of clinical consultations. The work will generate large amounts of rich data for analysis and we feel that inclusion of more patients would be unmanageable and would entail sacrificing depth of analysis. We are also engaged in ethnographic observation of professional activities, professional interviews and video-reflexive research workshops (which are recorded as data). We have included a reference to these professional participants (40 to date, and recruitment is ongoing) in our section entitled Setting and Context as this was an unintended omission in our original manuscript.

We have explicitly acknowledged the over-representation of non-white Caucasian participants in the manuscript as a limitation. This arises primarily from the requirements that the participants can speak adequate English and are aged 65+. In the urban, multi-ethnic area in which we are recruiting patients (Site A and C) the older patients rarely speak any English. Our methodological approach is highly dependent on analysing talk, text and interactions. We do not have the skills within the research team to accommodate the complexity that multiple languages would bring (and neither do we have the necessary research budget). We are conducting a smaller study separately that includes a bilingual researcher by way of addressing this limitation.

Methods: It was mentioned that a purposive sampling approach was used, but information about data saturation is lacking.

This is a study protocol and our analysis is at an early stage. We are unable to comment on data saturation as this point.

Methods: A more detailed data analysis section is missing in this protocol paper. I assume that it is difficult as this protocol consists of different parts. Please elaborate.

An earlier version of our submitted manuscript included a methods section and a separate analysis section in several parts (to reflect the diverse work streams of the project) but we found this unsatisfactory and thought it was likely to be very confusing to readers. We therefore decided to integrate our approach to analysis with our explanation of methods (e.g. the 'sociology of routines' literature which we outline in the section called "Professional experiences and practices" and the references to 'practice theory' which is an important overarching analytical orientation across our work. We will of course also draw on other relevant social theory as our analysis advances, but are unable at this stage to set out precisely what these will be as we are very early in our analysis work. It is likely that we will draw on the 'Burden of Treatment' theory and have included reference to this (in the section on Integrating Analysis) by way of providing one example of potentially relevant theory.

VERSION 2 – REVIEW

REVIEWER	Dr. Wilma Knol University Medical Center Utrecht The Netherlands
REVIEW RETURNED	24-Jul-2019
GENERAL COMMENTS	I would like to thank the authors. They responded adequately to the comments and adapted their manuscript accordingly.